# *Helicobacter pylori* Eradication Does Not Adversely Affect the Clinical Course of Gastric Cancer: A Multicenter Study on Screening Endoscopic Examination in Japan

**DOI:** 10.3390/cancers16040733

**Published:** 2024-02-09

**Authors:** So Takahashi, Kenta Watanabe, Sho Fukuda, Tatsuki Yoshida, Takahiro Dohmen, Junichi Fujiwara, Mari Matsuyama, Shusei Fujimori, Masato Funaoka, Kodai Shirayama, Yohei Horikawa, Saki Fushimi, Shu Uchikoshi, Kengo Onochi, Ryo Okubo, Takao Hoshino, Toru Horii, Taira Kuramitsu, Kotaro Sakaki, Toru Ishii, Taiga Komatsu, Yuko Yoshida, Kenji Shirane, Tsuyoshi Ono, Yosuke Shimodaira, Tamotsu Matsuhashi, Katsunori Iijima

**Affiliations:** 1Department of Gastroenterology, Akita University Graduate School of Medicine, 1-1-1 Hondo, Akita 010-8543, Akita, Japan; 2Department of Gastroenterology, Yuri Kumiai General Hospital, 38 Ienoushiro, Kawaguchi, Yurihonjo 015-8511, Akita, Japan; 3Department of Gastroenterology, Yokote Municipal Hospital, 5-31 Negishi, Yokote 013-8602, Akita, Japan; 4Department of Gastroenterology, Hiraka General Hospital, 3-1 Yatsukuchi, Maego, Yokote 013-8610, Akita, Japan; hrkjmsom@air.ocn.ne.jp (Y.H.);; 5Department of Gastroenterology, Omagari Kosei Medical Center, 8-65 Omagaritori, Daisen 014-0027, Akita, Japan; 6Department of Gastroenterology, Akita Kosei Medical Center, 1-1-1 Nishibukuro, Iijima, Akita 011-0948, Akita, Japan; 7Department of Gastroenterology, Akita Red Cross Hospital, 222-1 Nawashirosawa, Kamikitatesaruta, Akita 010-1495, Akita, Japan; 8Department of Gastroenterology, Honjo-Daiichi Hospital, 110 Iwabuchishita, Yurihonjo 015-8567, Akita, Japan; 9Department of Gastroenterology, Shirane Hospital, 5-29 Kyokuhokusakae, Akita 010-0922, Akita, Japan; 10Department of Gastroenterology, Omori Municipal Hospital, 245-205 Sugouta, Omori, Yokote 013-0525, Akita, Japan

**Keywords:** gastric cancer, surveillance endoscopy, *Helicobacter pylori* eradication

## Abstract

**Simple Summary:**

Since gastric cancers (GCs) detected after *Helicobacter pylori* (*HP*) eradication present with different morphological characteristics from conventional *HP*-positive GCs, delayed detection of early-stage GCs may be observed. However, *HP* eradication is generally considered to be effective in preventing gastric cancer. Due to these contradictory facts, it remains unclear whether the impact of *HP* eradication on the clinical course of GC is truly beneficial. In this retrospective multicenter study conducted over 5 years, a total of 231 patients with GCs were newly diagnosed and enrolled exclusively through screening endoscopy. Propensity analysis showed that *HP* eradication was not significantly associated with deeper tumor invasion. *HP* eradication did not lead to delayed diagnosis of GCs, supporting the recommendation of *HP* eradication in screening programs to reduce the total number of GC cases.

**Abstract:**

Background: Since gastric cancers (GCs) detected after *Helicobacter pylori* (*HP*) eradication present with different morphological characteristics from conventional *HP*-positive GCs, delayed detection of early-stage GCs may be observed. This study aimed to investigate the clinical impact of *HP* eradication on diagnosing GC during screening endoscopy. Methods: Eleven health checkup institutions in Japan participated in the present study. All GC cases newly diagnosed by screening endoscopy between January 2016 and December 2020 were included. After propensity score matching, multivariable regression analysis was performed to estimate the effect of *HP* eradication on deep tumor invasion among *HP*-eradicated and *HP*-positive GC cases. Results: A total of 231 patients with GCs (134 *HP*-eradicated and 97 *HP*-positive cases) were enrolled. After propensity score matching, there were 81 cases in each group. The distribution of the depth of tumor invasion (pT1a, pT1b1, pT1b2, and pT2) between the *HP*-eradicated group and *HP*-positive group was similar (*p* = 0.82). In the propensity analysis, with *HP*-positive as the reference value, *HP* eradication was not significantly associated with T1b–T4-GCs and T1b2–T4-GCs, with odds ratios (95% confidence intervals) of 1.16 (0.48–2.81) and 1.16 (0.42–3.19), respectively. Conclusions: *HP* eradication does not adversely affect the clinical course of GCs, supporting the recommendation of *HP* eradication in screening programs to reduce the total number of GC cases without delaying diagnosis.

## 1. Introduction

Though the incidence and mortality rates of gastric cancer (GC) have been decreasing in Japan, GC continues to be one of the most frequently diagnosed cancers [1]. *Helicobacter pylori* (*HP*) is the main cause of GC, and eradication therapy is widely performed in Japan. Although the risk of developing GC decreases with eradication therapy—a recent meta-analysis showed a 46% reduction [2]—it may persist even after a successful *HP* eradication [3,4,5]. In fact, the proportion of GCs detected after *HP* eradication has been gradually increasing in Japan [6].

GCs detected after *HP* eradication present with different morphological characteristics from conventional *HP*-positive GCs. For example, GCs that developed after *HP* eradication may appear with a gastritis-like appearance or may be covered with non-neoplastic epithelium [7,8,9,10,11,12], which may hinder the detection of early-stage GC, potentially leading to delayed detection of GC after eradication. However, previous studies included only patients with early-stage GC. Hence, the clinical impact of *HP* eradication on the entire spectrum of GC, including screening findings of both early-stage and advanced-stage GCs, remains unclear. We considered that the overall impact of *HP* eradication on the clinical course of GC (e.g., depth or treatment choice) should be examined in an unselected GC cohort by including the entirety of the cases detected.

In the Akita Prefecture, where the incidence of GC is highest in Japan [13,14,15], several facilities have been offering screening endoscopies as part of their health checkup. Using data collected from multiple institutions, this study aimed to examine whether *HP* eradication is associated with a delayed diagnosis of GC (e.g., GC with deeper tumor invasion) by comparing the clinical stage at diagnosis of *HP*-positive and *HP*-eradicated individuals through screening endoscopy.

## 2. Patients and Methods

### 2.1. Patients

A multi-institutional, retrospective observational study involving 11 institutions in Akita Prefecture, Japan that perform screening endoscopies as part of a health checkup was conducted. All patients newly diagnosed with GC through screening endoscopy at these institutions between January 2016 and December 2020 were included in this study. Patients that were either *HP*-positive or *HP*-eradicated were defined as eligible participants. Those with GCs that occurred in the remnant stomach or lacked final pathological findings were excluded from this study.

### 2.2. Definition of HP Infection Status of GC

*HP*-positive GCs were defined as those with *HP* infection at the time of diagnosis, and *HP*-eradicated GCs were defined as those who were diagnosed at least one year after successful *HP* eradication. A patient was considered to be *HP*-positive if the patient tested positive in at least one of the following without a history of eradication therapy: A serum antibody test, stool antigen test, ^13^C-urea breath test, rapid urease test, or histological examination. Successful eradication was judged by a negative ^13^C-urea breath test [16].

### 2.3. Data Collection

Medical records were retrospectively reviewed to retrieve data regarding information at the time of GC diagnosis, such as patients’ sex, age, history of endoscopic resection for early-stage GC (primary or metachronous GC), tobacco and alcohol consumption, and the interval between previous and diagnostic endoscopies. In Japan, a two-year interval is recommended for screening endoscopy to detect early-stage GCs [17]. Therefore, the intervals between previous and diagnostic endoscopy examinations were defined as ≤1 year, >1 year to ≤2 years, and >2 years or never undergone. As cancer lesion-related factors, data on the longitudinal location (upper, middle, or lower part of the stomach) of the tumor, macroscopic type (elevated or depressed), tumor size (<20 mm or ≥20 mm), and histological type (differentiated or undifferentiated) were also collected. Data on the depth of tumor invasion were retrieved from the final pathological reports. The review was conducted by more than two experts with >10 years of experience.

### 2.4. Statistical Analysis

Categorical variables were expressed as the number and proportion (%) and compared using a Chi-square test or Fisher’s exact test, as appropriate. Continuous variables were expressed as the median with interquartile range and compared using a Mann–Whitney U test. Propensity score matching was applied to reduce selection bias and adjust for significant differences in baseline clinical characteristics between the groups. A propensity score was calculated using a logistic regression model incorporating the patients’ age, sex, smoking status, drinking status, and the presence of either a primary or metachronous GC as independent variables. The *HP*-positive and *HP*-eradicated groups were matched in a 1:1 ratio using the nearest neighbor method with a caliper width of 0.2 of the standard deviation of the logit of the propensity score. The effect of matching was assessed by an absolute standardized difference (ASD), and a covariate of ASD ≤ 0.1 was considered to be balanced. After propensity score matching, the association between *HP* eradication and GCs with deep tumor invasion was estimated using logistic regression analysis. Additionally, other factors that can influence the development of GCs with deep tumor invasion were analyzed in another multivariable model. The results of the regression analyses were expressed using odds ratios (ORs) and 95% confidence intervals (CIs). All statistical analyses were conducted using the EZR version 1.63 (Saitama Medical Center, Jichi Medical University, Saitama, Japan) [18], and a *p* < 0.05 was considered statistically significant.

### 2.5. Ethical Statement

All procedures were performed in accordance with the Helsinki Declaration of 1964 and later versions. The study protocol was approved by the Ethics Committee of Akita University (ID: 2849) and each participating institute. The need for informed consent was waived by the corresponding Ethics Committees because of the retrospective nature of this study.

## 3. Results

### 3.1. Characteristics of Study Population before and after Propensity Score Matching

During the study period, 141,621 screening esophagogastroduodenoscopies (EGDs) were performed in 11 health checkup institutions. From these, 296 (0.21%) patients were diagnosed with GC. The GC detection rate was similar to that reported for health checkup endoscopy in another area of Japan (0.22%) [19]. Among the 296 patients, 11 patients had final pathological results that could not be tracked, 6 patients had GCs arising in the remnant stomach, 39 were *HP*-undetected cases without a history of eradication, and 9 were *HP*-unknown cases. These patients were excluded from the study. A total of 231 cases of GC (97 *HP*-positive and 134 *HP*-eradicated cases) were included in this study (Figure 1). In terms of tumor invasion depth, there were 168 (72.7%) cases of pT1a, 14 (6.1%) cases of pT1b1, 33 (14.3%) cases of pT1b2, and 16 (6.9%) cases of pT2 or deeper. In terms of treatment, 165 patients (71.4%) underwent endoscopic submucosal dissection (ESD), 64 patients (27.7%) underwent surgery, 1 patient (0.4%) underwent chemotherapy, and 1 patient (0.4%) did not undergo any treatment. Table 1 shows the clinical characteristics of the patients, before and after propensity score matching. Before propensity score matching, the patients’ median age was 65 years (60–70), 199 (86.2%) were male, and 18 (7.8%) had metachronous GC. After propensity score matching, the patients’ median age was 64.5 years (60–70), 143 (88.3%) were male, and 12 (7.4%) had metachronous GC. All five matching factors were considered well-balanced. Both before and after propensity score matching, the interval between previous and diagnostic endoscopic examination was significantly longer in the *HP*-positive group than in the *HP*-eradicated group (*p* < 0.01). Although there was no significant difference in the macroscopic tumor type and tumor size before matching, depressed tumors and tumors measuring <20 mm were significantly more common in the *HP*-eradicated group after matching (*p* = 0.05 and *p* = 0.01, respectively). There was no significant difference in the depth of tumor invasion (pT1a, pT1b1, pT1b2, and pT2 or deeper) between the *HP*-eradicated group and the *HP*-positive group both before and after propensity score matching (71.6%, 6.7%, 14.9%, and 6.7% vs. 74.2%, 5.2%, 13.4%, and 7.2%, *p* = 0.96; 74.1%, 6.2%, 14.8%, and 4.9% vs. 69.1%, 6.2%, 16.0%, and 8.6%, *p* = 0.82).

### 3.2. Association between HP Eradication and GCs with Deep Invasion

Among 162 propensity score-matched GC cases, multivariable regression analysis was performed to estimate the association between *HP* eradication and GCs with pT1b or deeper invasion. As a result, the adjusted OR (95% CI) for *HP* eradication was 1.16 (0.48–2.81), suggesting no significant association with GCs with pT1b or deeper invasion (Table 2). Similarly, regarding the association of GCs with pT1b2 or deeper invasion, the adjusted OR (95% CI) for *HP* eradication was 1.16 (0.42–3.19), suggesting no significant association (Table 3). In sensitivity analyses, where *HP*-eradicated GCs were defined as those who were diagnosed at least two years after successful eradication, the results were largely unchanged, i.e., *HP* eradication was not significantly associated with deeper tumor invasion (Appendix A).

### 3.3. Estimating Factors Associated with GCs with Deep Invasion

To estimate the associations between factors other than *HP* eradication, multivariable regression analysis was performed without propensity score matching. Factors associated with the development of GCs with pT1b or deeper invasion were its location (if in the upper third of the stomach), a tumor size ≥20 mm, and an undifferentiated histological type (Table 4). Similarly, factors associated with the development of GCs with pT1b2 or deeper invasion were being male, its location (if in the upper third of the stomach), a tumor size ≥20 mm, an undifferentiated histological type, or if the interval between previous and diagnostic endoscopy was >2 years or if it was never undergone (Table 5). Again, *HP* eradication was not associated with GCs with deeper tumor invasion in both multivariable models. Furthermore, a comparison between early-stage and advanced-stage GC revealed that although advanced-stage GC was more likely to have a size ≥20 mm and be of the undifferentiated type, once again *HP* eradication was not significantly associated with advanced-stage GC (Appendix A).

## 4. Discussion

This study demonstrated that *HP* eradication does not significantly impact the depth of tumor invasion in GCs detected during screening EGD. Findings clinically significant to the depth of tumor invasion, such as the use of ESD for pT1a–pT1b1, and surgery for pT1b2 or deeper, are directly related to the recommended treatment.

Because healthcare insurance for eradication therapy targeting all *HP*-positive individuals has been widely implemented in Japan since 2013 [20], a substantial proportion of *HP*-positive patients have received treatment. With the increased number of *HP*-eradicated individuals, an increasing number of GCs are being diagnosed in these patient groups, as GCs still emerge even after successful *HP* eradication [5,21,22]. Consequently, recent studies have reported that the number of diagnosed GCs among patients that are *HP*-eradicated has already exceeded that in those that are *HP*-positive [6,23], which holds true in the current study. Therefore, GC detection by EGD in eradicated individuals, compared to traditionally prevalent *HP*-positive GC, has great clinical implications.

Studies have shown that *HP* eradication conceals the findings of early cancer lesions, which may lead to a delayed cancer diagnosis [7,8,9,10,24]. Moreover, two previous studies demonstrated that this concern could be a real issue by showing that post-eradicated GC showed a higher proportion of submucosal invasion than cases of *HP*-positive GC [11,12]. However, these two studies suffered from major selection bias, as the authors admitted, since the studies comprised GC patients who received ESD, excluding cases with early GC beyond the indications for ESD and those with advanced GC. In addition, these studies were performed in high-volume centers, which may have caused a referral bias [11,12]. To minimize this bias, the current study included an entire GC cohort consisting of those diagnosed via screening EGD in multiple health checkup institutes.

Propensity analysis demonstrated that *HP* eradication did not affect GCs with pT1b or deeper. Meanwhile, consistent with previous reports, we observed some morphological differences in GC between *HP*-eradicated and *HP*-positive subjects, e.g., the prevalence of the depressed type and small GCs was higher among patients that are *HP*-eradicated than in those that are *HP*-positive, after propensity score matching [8,10,11,25]. Still, *HP* eradication was not significantly associated with GCs with deep tumor invasion, even after adjusting for such cancer lesion-related factors, suggesting that the effect of *HP* eradication does not predispose one to develop GC with deep tumor invasion. As studies have demonstrated, *HP* eradication may make it difficult to diagnose early-stage GC by obscuring the characteristic lesions [7,8,9,10,24]. At the same time, other studies demonstrated that *HP* eradication decreased the proliferation of GC [7,25], suggesting that the growth of neoplastic lesions was decelerated in *HP*-eradicated cases [21,26,27,28]. Hence, the potentially delayed diagnosis of GCs after *HP* eradication due to the difficulty in endoscopic detection will be counterbalanced by the relatively slow growth of the cancer lesions. Consequently, *HP* eradication may yield no substantial disadvantages for detecting GC at an early stage, as demonstrated in the current study.

In the current study, multivariable regression models without propensity score matching revealed that tumors located in the upper third of the stomach, measuring ≥20 mm, and with an undifferentiated histological type are highly associated with T1b or deeper GCs. In addition to these cancer lesion-related factors, when the subjects of interest were defined as GCs with pT1b2 or deeper, a long-term interval (>two years or never undergone) between previous and diagnostic endoscopic examination was found to be significantly associated with deeper tumor invasion (Table 5). Screening EGD for GC performed every two years, which is the current recommendation in Japan [17,29,30], may be effective in detecting GCs before pT1b2 or deeper invasion, which requires surgical resection.

This study minimized selection bias by including consecutive patients with GCs of different stages diagnosed through screening EGD exclusively from 11 health checkup institutions over 5-year period. Thus, it provides a comprehensive report on the occurrence of GC in the area. Furthermore, this study was conducted in Akita Prefecture, Japan, where the mortality from GC is most rampant, and urgent actions are needed [13,14,15].

Attention should be paid to several limitations associated with this study. First, the retrospective nature of this study may be a limitation of this study, although each GC case was systematically registered at each institute once it was diagnosed at health checkup. Second, morphological and histological changes over time after successful eradication could not be evaluated in this study. Third, despite adopting a multicenter study design over five years, there were only 16 cases of advanced-stage GC, which limited the ability to conduct sufficient subgroup analysis. Finally, some study patients may have been misclassified regarding *HP* infection status. In this study, possible reinfection after successful eradication may have occurred, although the frequency of reinfection should be very low once the success was confirmed using a ^13^C-urea breath test [31]. In addition, in patients with a positive serum antibody test, *HP* may have been spontaneously eliminated due to advanced gastric atrophy, although such cases are supposed to be less common [32].

## 5. Conclusions

In conclusion, *HP* eradication does not adversely affect the clinical course of GC patients. Together with the well-established fact that *HP* eradication reduces the risk of developing GC [2], the findings in this study support the recommendation of *HP* eradication in screening programs to reduce the total number of GC cases without delaying diagnosis.

## Figures and Tables

**Figure 1 cancers-16-00733-f001:**
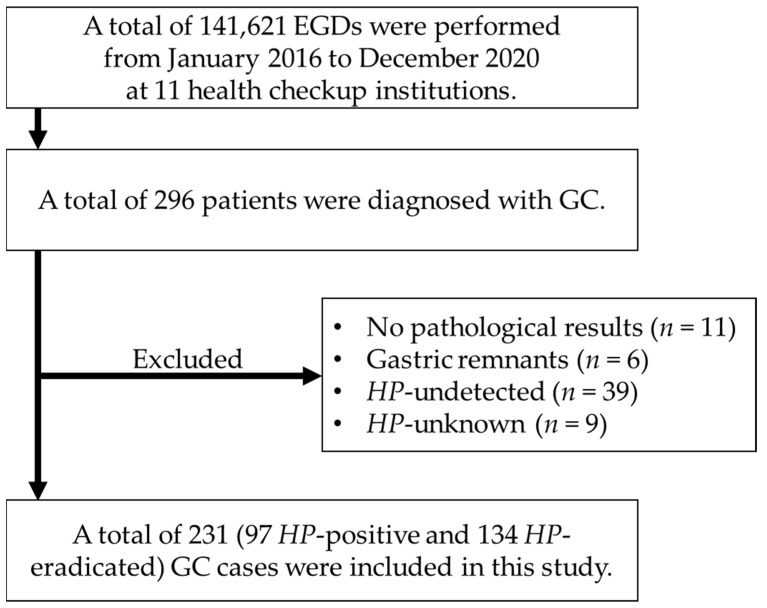
Flow diagram of the enrollment of this study. *EGD* esophagogastroduodenoscopy; *GC* gastric cancer; *HP Helicobacter pylori*.

**Table 1 cancers-16-00733-t001:** Baseline characteristics of the study patients before and after propensity score matching.

	Before Propensity Score Matching				After Propensity Score Matching			
	*HP*-Eradicated (*n* = 134)	*HP*-Positive (*n* = 97)	*p* Value	ASD	*HP*-Eradicated (*n* = 81)	*HP*-Positive (*n* = 81)	*p* Value	ASD
Male sex, *n* (%)	115 (85.8)	84 (86.6)	1.00	0.02	72 (88.9)	71 (87.7)	1.00	0.04
Age, years, median (IQR)	65 (61–71)	64 (59–70)	0.31	0.13	64 (59–71)	66 (60–70)	0.94	0.03
Metachronous cancer, *n* (%)	11 (8.2)	7 (7.2)	1.00	0.04	6 (7.4)	6 (7.4)	1.00	<0.01
Smoking status, *n* (%)								
Current smoker	35 (26.1)	28 (28.9)	0.84	0.08	22 (27.2)	24 (29.6)	0.89	0.07
Past smoker	27 (20.1)	17 (17.5)			16 (19.8)	14 (17.3)		
Never smoker	67 (50.0)	48 (49.5)			43 (53.1)	43 (53.1)		
Unknown	5 (3.7)	4 (4.1)			0 (0.0)	0 (0.0)		
Drinking status, *n* (%)								
Current drinker	92 (68.7)	68 (70.1)	1.00	0.04	60 (74.1)	61 (75.3)	1.00	0.07
Past drinker	5 (3.7)	3 (3.1)			3 (3.7)	2 (2.5)		
Never drinker	30 (22.4)	21 (21.6)			18 (22.2)	18 (22.2)		
Unknown	7 (5.2)	5 (5.2)			0 (0.0)	0 (0.0)		
Longitudinal location, *n* (%)								
Upper third	28 (20.9)	15 (15.5)	0.31	0.14	16 (19.8)	14 (17.3)	0.84	0.06
Middle or lower third	106 (79.1)	82 (84.5)			65 (80.2)	67 (82.7)		
Macroscopic type, *n* (%)								
Elevated type	22 (16.4)	24 (24.7)	0.13	0.21	10 (12.3)	21 (25.9)	0.05	0.35
Depressed type	112 (83.6)	73 (75.3)			71 (87.7)	60 (74.1)		
Tumor size, *n* (%)								
<20 mm	90 (67.2)	55 (56.7)	0.13	0.22	59 (72.8)	43 (53.1)	0.01	0.42
≥20 mm	44 (32.8)	42 (43.3)			22 (27.2)	38 (46.9)		
Histological type, *n* (%)								
Differentiated type	112 (83.6)	83 (85.6)	0.72	0.06	71 (87.7)	70 (86.4)	1.00	0.04
Undifferentiated type	22 (16.4)	14 (14.4)			10 (12.3)	11 (13.6)		
Interval between previous and diagnostic endoscopic examination, *n* (%)								
≤1 year	78 (58.2)	30 (30.9)	<0.01	0.73	45 (55.6)	25 (30.9)	<0.01	0.75
>1 year, ≤2 years	28 (20.9)	15 (15.5)			21 (25.9)	14 (17.3)		
>2 years or never	28 (20.9)	52 (53.6)			15 (18.5)	42 (51.9)		
Depth of tumor invasion, *n* (%)								
T1a	96 (71.6)	72 (74.2)	0.96	0.08	60 (74.1)	56 (69.1)	0.82	0.16
T1b1	9 (6.7)	5 (5.2)			5 (6.2)	5 (6.2)		
T1b2	20 (14.9)	13 (13.4)			12 (14.8)	13 (16.0)		
T2 or deeper	9 (6.7)	7 (7.2)			4 (4.9)	7 (8.6)		

*ASD* absolute standardized difference; *HP Helicobacter pylori*; *IQR* interquartile range.

**Table 2 cancers-16-00733-t002:** Risk of T1b–T4-GC associated with *HP* eradication.

		Adjusted OR ^a^	95% CI	*p* Value
Longitudinal location	Upper third	2.01	0.78–5.13	0.15
	Middle or lower third	reference		
Macroscopic type	Depressed type	2.64	0.77–9.08	0.12
	Elevated type	reference		
Tumor size	≥20 mm	5.13	2.30–11.40	<0.01
Histological type	Undifferentiated type	3.23	1.11–9.37	0.03
	Differentiated type	reference		
Interval between previous and diagnostic endoscopic examination	≤1 year	reference		
	>1 year, ≤2 years	1.32	0.45–3.90	0.62
	>2 years or never	1.76	0.68–4.58	0.25
*HP* infection status	*HP*-eradicated	1.16	0.48–2.81	0.74
	*HP*-positive	reference		

*CI* confidence interval; *GC* gastric cancer; *HP Helicobacter pylori*; *OR* odds ratio; ^a^ Adjusted by longitudinal location, macroscopic type, tumor size, histological type, interval between previous and diagnostic endoscopic examination, and *HP* infection status.

**Table 3 cancers-16-00733-t003:** Risk of T1b2–T4-GC associated with *HP* eradication.

		Adjusted OR ^a^	95% CI	*p* Value
Longitudinal location	Upper third	3.16	1.10–9.05	0.03
	Middle or lower third	reference		
Macroscopic type	Depressed type	11.10	1.30–94.00	0.03
	Elevated type	reference		
Tumor size	≥20 mm	7.15	2.81–18.20	<0.01
Histological type	Undifferentiated type	2.98	0.96–9.31	0.06
	Differentiated type	reference		
Interval between previous and diagnostic endoscopic examination	≤1 year	reference		
	>1 year, ≤2 years	2.55	0.72–9.08	0.15
	>2 years or never	2.71	0.88–8.33	0.08
*HP* infection status	*HP*-eradicated	1.16	0.42–3.19	0.78
	*HP*-positive	reference		

*CI* confidence interval; *GC* gastric cancer; *HP Helicobacter pylori*; *OR* odds ratio; ^a^ Adjusted by longitudinal location, macroscopic type, tumor size, histological type, interval between previous and diagnostic endoscopic examination, and *HP* infection status.

**Table 4 cancers-16-00733-t004:** Estimating factors associated with T1b–T4-GC.

		Adjusted OR ^a^	95% CI	*p* Value
Sex	Male	1.58	0.54–4.63	0.40
Age		0.98	0.94–1.03	0.43
Metachronous cancer		0.19	0.02–1.60	0.13
Longitudinal location	Upper third	2.36	1.06–5.25	0.04
	Middle or lower third	reference		
Macroscopic type	Depressed type	1.27	0.49–3.29	0.63
	Elevated type	reference		
Tumor size	≥20 mm	5.88	2.95–11.70	<0.01
Histological type	Undifferentiated type	3.35	1.43–7.86	<0.01
	Differentiated type	reference		
Interval between previous and diagnostic endoscopic examination	≤1 year	reference		
	>1 year, ≤2 years	1.16	0.44–3.05	0.77
	>2 years or never	1.81	0.81–4.07	0.15
*HP* infection status	*HP*-eradicated	1.63	0.77–3.44	0.20
	*HP*-positive	reference		

*CI* confidence interval; *GC* gastric cancer; *HP Helicobacter pylori*; *OR* odds ratio; ^a^ Adjusted by sex, age, metachronous cancer, longitudinal location, macroscopic type, tumor size, histological type, interval between previous and diagnostic endoscopic examination, and *HP* infection status.

**Table 5 cancers-16-00733-t005:** Estimating factors associated with T1b2–T4-GC.

		Adjusted OR ^a^	95% CI	*p* Value
Sex	Male	4.75	1.07–21.20	0.04
Age		0.97	0.92–1.02	0.29
Metachronous cancer		0.31	0.03–2.77	0.29
Longitudinal location	Upper third	2.70	1.12–6.47	0.03
	Middle or lower third	reference		
Macroscopic type	Depressed type	1.80	0.57–5.71	0.32
	Elevated type	reference		
Tumor size	≥20 mm	7.92	3.56–17.60	<0.01
Histological type	Undifferentiated type	3.71	1.48–9.33	<0.01
	Differentiated type	reference		
Interval between previous and diagnostic endoscopic examination	≤1 year	reference		
	>1 year, ≤2 years	1.84	0.63–5.36	0.27
	>2 years or never	2.76	1.09–6.96	0.03
*HP* infection status	*HP*-eradicated	1.74	0.75–4.03	0.20
	*HP*-positive	reference		

*CI* confidence interval; *GC* gastric cancer; *HP Helicobacter pylori*; *OR* odds ratio; ^a^ Adjusted by sex, age, metachronous cancer, longitudinal location, macroscopic type, tumor size, histological type, interval between previous and diagnostic endoscopic examination, and *HP* infection status.

## Data Availability

Data presented in this study are available upon reasonable request from the corresponding authors.

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
