# Peer review of "Helicobacter pylori Eradication Does Not Adversely Affect the Clinical Course of Gastric Cancer: A Multicenter Study on Screening Endoscopic Examination in Japan"

_cancers, 2024, doi:10.3390/cancers16040733_

Round 1

Reviewer 1 Report

Comments and Suggestions for Authors

This is an interesting multicenter study on gastroscopy screening which concluded that Hp eradication does not adversely affect the clinical course of gastric cancer. Followings are my suggestions to improve the study:

Major comments

1. Main flaw of this study would be the lack of Hp eradication period before the screening. Morphological and histological changes after Hp eradication cannot be assessed only by comparing Hp-infected and past-infected groups. To see the difference, time of Hp eradication is important.

2. Another limitation is the small number of AGC patients. Following the 2nd paragraph of Introduction, focus more on comparing EGC and AGC instead of comparing EGC T1a, T1b1, and T1b2.

3. Introduction (lines 69-71) “The clinical impact of HP eradication on the entire spectrum of GC, including screening findings of both early-stage and advanced-stage GCs, remains unclear”. - Add study hypothesis. What did the authors expected as a different morphological/histological change between the EGCs and AGCs after Hp eradication?

4. Methods - There are only 16 AGC patients among the 231 GC patients in this study; hence, additional analysis is needed to see the difference between EGC and AGC patients. Propensity score matching should be considered thereafter.

5. Results In addition to comment 1, add data on eradication period. To see mucosal changes after Hp eradication, time of successful Hp eradication is more important than gastroscopy interval.

6. Discussion - Compare the AGC and EGC data, and explain why Hp eradication did not delay the diagnosis by changing the morphology in both groups (which is different from references 7–10 and 24). Also, explain why Hp eradication did not led to cancer invasion to the deeper level in this study.

7. Conclusions - On propensity score matching, there was no significance between the 81 GC patients and 81 controls in this study. Unfortanetly, there is no statistical evidence that the study results support the recommendation of Hp eradication in screening programs to reduce GCs. In other words, morphological changes after Hp eradication did not delay the diagnosis of EGC during the screening, but this does not indicate recommendation for Hp eradication. To conclude like this, superiority of the Hp-eradicated group should be added in the Results.

Minor comments

1. Title (line 2) - There is a typo. Change Dose to Does.

2. Abstract (line 41) - Study aim is missing. Describe that the authors tried to determine clinical impact of Hp eradication on EGCs detected during gastroscopy screening.

Comments on the Quality of English Language

Typos should be revised.

Author Response

This is an interesting multicenter study on gastroscopy screening which concluded that Hp eradication does not adversely affect the clinical course of gastric cancer. Followings are my suggestions to improve the study:

-We appreciate your favorable and encouraging comments. We endeavor to address all the comments satisfactorily.

Major comments

  1. Main flaw of this study would be the lack of Hp eradication period before the screening. Morphological and histological changes after Hp eradication cannot be assessed only by comparing Hp-infected and past-infected groups. To see the difference, time of Hp eradication is important.

-We appreciate your insightful comment. As you pointed out, we agree that morphological and histological changes over time after eradication are important to characterize the features of GC after eradication. Unfortunately, we could not evaluate morphological or histological changes before and after eradication in individual cases. We acknowledge that this is a limitation of the present study. Instead, in this study, we took another approach to characterize the features of GC after eradication by comparing the clinical features, including the cancer depth and treatment, between HP-positive and eradicated patients. We believe that it is practical approach to understand the clinical characteristics of GC after eradication. We added the tables and descriptions (Table S1–S3, lines 182–185 and 279–280).

  1. Another limitation is the small number of AGC patients. Following the 2ndparagraph of Introduction, focus more on comparing EGC and AGC instead of comparing EGC T1a, T1b1, and T1b2.

-We appreciate your insightful comment. As we have already mentioned, previous studies included only EGC patients, and the clinical impact of HP eradication on the entire spectrum GC remains unclear. Unfortunately, there were only 16 AGC cases in this study, making it difficult to conduct a thorough analysis. We added this as a limitation (lines 280–282). Nevertheless, we did our best to perform an additional analysis regarding EGC vs AGC (Table S4).

  1. Introduction (lines 69-71) “The clinical impact of HP eradication on the entire spectrum of GC, including screening findings of both early-stage and advanced-stage GCs, remains unclear”. - Add study hypothesis. What did the authors expected as a different morphological/histological change between the EGCs and AGCs after Hp eradication?

-We appreciate your insightful comment. Previous studies have limited their cases to GC with indication of endoscopic resection. Therefore, the impact of eradication in patients including GC beyond the indication of endoscopic resection remains unclear. We added our premise of this study in the revision (lines 73–75).

  1. Methods - There are only 16 AGC patients among the 231 GC patients in this study; hence, additional analysis is needed to see the difference between EGC and AGC patients. Propensity score matching should be considered thereafter.

-We appreciate your constructive comment. Unfortunately, there were only 16 AGC cases in this study, making it difficult to conduct a thorough analysis. We added this as a limitation. However, as you pointed out, we also recognize the importance of comparing EGC and AGC. Therefore, we conducted an additional analysis regarding EGC vs AGC and there was no difference in HP infection status between the two groups. We added the table and the corresponding description (Table S4 and lines 207–209).

  1. Results In addition to comment 1, add data on eradication period.To see mucosal changes after Hp eradication, time of successful Hp eradication is more important than gastroscopy interval.

-We appreciate your constructive comment. As we mentioned above, we could not assess mucosal changes after eradication. This was added as a limitation in this study. Instead, regarding time course after eradication, we conducted additional sensitivity analyses in which the eradicated GCs were defined as those who were diagnosed at least two years after successful eradication instead of at least 1 year. Consequently, along with the original results, successful eradication was not significantly associated with deeper tumor invasion in those analyses (Table S1–S3 and lines 182–185).

  1. Discussion - Compare the AGC and EGC data, and explain why Hp eradication did not delay the diagnosis by changing the morphology in both groups (which is different from references 7–10 and 24). Also, explain why Hp eradication did not led to cancer invasion to the deeper level in this study.

-We appreciate your insightful comment. As you indicated, our finding that eradication does not lead to deeper levels of tumor invasion is different from previous studies. We consider the difference comes from the study design. In the previous studies, the investigators mainly focused on the early GC with indication of endoscopic resection in high-volume centers, which could have caused selection and referral biases. In contrast, to minimize these biases, the current study included an entire GC cohort consisting of those diagnosed via screening EGD in multiple health checkup institutes. To explain the difference in the study design between previous and our studies, we added the description to the revised version (lines 240–243). In addition, regarding why HP eradication did not lead to cancer invasion to the deeper level in this study, we speculate that HP eradication decreased the proliferation of GC, leading to deceleration of the growth of neoplastic lesions in eradicated cases as we already mentioned in the initial version (lines 253–255). To show our thoughts more clearly, we added the description regarding these (lines 255–257).

  1. Conclusions - On propensity score matching, there was no significance between the 81 GC patients and 81 controls in this study. Unfortanetly, there is no statistical evidence that the study results support the recommendation of Hp eradication in screening programs to reduce GCs. In other words, morphological changes after Hp eradication did not delay the diagnosis of EGC during the screening, but this does not indicate recommendation for Hp eradication. To conclude like this, superiority of the Hp-eradicated group should be added in the Results.

-Thank you for your constructive comments. In this regard, two points need to be recognized to guide our conclusion. The first point is the well-established fact that HP eradication reduces the risk of developing GC (Gut 2020;69:2113–2121). The second point is the fact demonstrated in this multicenter study that HP eradication is not associated with deep tumor invasion. That is, compared with HP-positive GC, HP-eradicated GC is not associated with deep tumor invasion. We believe that these two points support the recommendation of eradication in screening program. As you pointed out, this conclusion cannot be drawn based solely on the results of this study. We recognize that a prerequisite knowledge is required that eradication reduces GC risk. We revised the description regarding those (lines 291–294).

Minor comments

  1. Title (line 2) - There is a typo. Change Dose to Does.

-We appreciate your kind comment. We have corrected the title (line 2).

  1. Abstract (line 41) - Study aim is missing. Describe that the authors tried to determine clinical impact of Hp eradication on EGCs detected during gastroscopy screening.

-We appreciate your kind comment. We revised the abstract according to your advice (lines 41–42).

Reviewer 2 Report

Comments and Suggestions for Authors

The paper is interesting and well written. I have only few comments:

- It would be helpful to know how the screening endoscopy is performed in the Akita prefecture: do you perform random biopsies -if no lesions are found- to assess Hp status or you just use non invasive methods to assess Hp? How often? If you do not use biopsy on random mucosa, how can be guaranteed that the patients diagnosed 1 year after the eradication were still free from the infection? Did they undergo parallel non-invasive screening? The retrospective design could lack of these information and some Hp negative patient could a false negative. Please, comment this point 

- Is the serum Ab test in Japan assessing IgM or IgG? And if it is assessing both, how can the authors be sure that the infection is still present instead of a recent clearence? Please, comment.

Comments on the Quality of English Language

The quality of English is good. Only minor editing is required 

Author Response

The paper is interesting and well written. I have only few comments:

-We appreciate your favorable and encouraging comment. We will endeavor to address all the comments satisfactorily.

- It would be helpful to know how the screening endoscopy is performed in the Akita prefecture: do you perform random biopsies -if no lesions are found- to assess Hp status or you just use non-invasive methods to assess Hp? How often? If you do not use biopsy on random mucosa, how can be guaranteed that the patients diagnosed 1 year after the eradication were still free from the infection? Did they undergo parallel non-invasive screening? The retrospective design could lack of these information and some Hp negative patient could a false negative. Please, comment this point.

-We appreciate your insightful comments. Because tests to detect HP infection, including random biopsies, were not routinely performed at the institutions participating in this study, we could not guarantee that there would be no recurrence of HP infection after successful eradication at the time of diagnostic endoscopy. However, given that eradication was accurately judged using a 13C-urea breath test in this study, the frequency of reinfection should be very low (J Gastroenterol 2012;47:641–646). Thus, we believe that this had minimum effect on the main results of this study. This was added to the manuscript as a limitation (lines 274–277).

- Is the serum Ab test in Japan assessing IgM or IgG? And if it is assessing both, how can the authors be sure that the infection is still present instead of a recent clearance? Please, comment.

-We appreciate your insightful comments. In Japan, only IgG is assessed, and 10 U/mL or more is considered HP-positive. As you pointed out, some patients with positive serum Ab test may have been eradicated spontaneously, although this is considered less common (APMIS 2003;111:619–624). We have added a limitation regarding possible misclassification of HP infection status (lines 277–279).

Round 2

Reviewer 1 Report

Comments and Suggestions for Authors

Thank you for submitting after revision. 

Although most of my comments were not resolved, the revised version is better than the original one.

Comments on the Quality of English Language

Editing is required for the revised parts. At least following sentences should be revised. 

1. Lines 181-183  In sensitivity analyses, where HP eradicated GCs were defined as those who were diagnosed at least two years after successful eradication, the results were largely unchanged ~.

2. Lines 206-208  Furthermore, a comparison of early-stage and advanced-stage GC revealed that although advanced-stage GC was more likely to be ≥20 mm and undifferentiated type, again, HP eradication was not significantly associated with advanced-stage GC (Table S4).

3. Lines 239-241  In addition, since these studies were performed in high-volume centers, which could have caused referral bias [11, 12].

4. Lines 254-256  Hence, potentially delayed diagnosis of GCs after HP eradication due to the difficulty in endoscopic detection will be counterbalanced by the relatively slow growth of the cancer lesions.

5. Lines 282-284  The possible reinfection after successful eradication may have occurred in this study, although the frequency of reinfection should be very low once the success was confirmed using 13C-urea breath test.

6. Lines 291-293 ~ the finding in this study supports the recommendation of HP eradication in screening programs to reduce the total number of GC cases without delaying the diagnosis.

Author Response

Thank you for submitting after revision. 

Although most of my comments were not resolved, the revised version is better than the original one.

- We sincerely appreciate your review of the revised version. Despite the limitations of the present study, we believe that your advice improved the paper.

Comments on the Quality of English Language

Editing is required for the revised parts. At least following sentences should be revised.

  1. Lines 181-183  In sensitivity analyses, where HP eradicated GCs were defined as those who were diagnosed at least two years after successful eradication, the results were largely unchanged ~.
  2. Lines 206-208  Furthermore, a comparison of early-stage and advanced-stage GC revealed that although advanced-stage GC was more likely to be ≥20 mm and undifferentiated type, again, HP eradication was not significantly associated with advanced-stage GC (Table S4).
  3. Lines 239-241  In addition, since these studies were performed in high-volume centers, which could have caused referral bias [11, 12].
  4. Lines 254-256  Hence, potentially delayed diagnosis of GCs after HP eradication due to the difficulty in endoscopic detection will be counterbalanced by the relatively slow growth of the cancer lesions.
  5. Lines 282-284  The possible reinfection after successful eradication may have occurred in this study, although the frequency of reinfection should be verylow once the success was confirmed using 13C-urea breath test.
  6. Lines 291-293 ~the finding in this study supports the recommendation of HP eradication in screening programs to reduce the total number of GC cases without delaying the diagnosis.

-We appreciate your comments. We have requested an external English proofreading service to revise the sentences you pointed out. Of these, Sentences 1 (lines 182–185) and 6 (lines 292–295) were judged not to require editing. The revised sentences are as follows.

  1. (Lines 182–185) In sensitivity analyses, where HP eradicated GCs were defined as those who were diagnosed at least two years after successful eradication, the results were largely unchanged, i.e. HP eradication was not significantly associated with deeper tumor invasion (Table S1–S3).
  2. (Lines 207–210) Furthermore, a comparison between early-stage and advanced-stage GC revealed that although advanced-stage GC was more likely to have a size ≥20 mm and be of undifferentiated type, once again, HP eradication was not significantly associated with advanced-stage GC (Table S4).
  3. (Lines 241–243) In addition, since these studies were performed in high-volume centers, which may have caused a referral bias [11, 12].
  4. (Lines 256–258) Hence, the potentially delayed diagnosis of GCs after HP eradication due to the difficulty in endoscopic detection will be counterbalanced by the relatively slow growth of the cancer lesions.
  5. (Lines 284–286) In this study, possible reinfection after successful eradication may have occurred, although the frequency of reinfection should be very low once the success was confirmed using 13C-urea breath test [31].
  6. (Lines 292–295) Together with the well-established fact that HP eradication reduces the risk of developing GC [2], the finding in this study supports the recommendation of HP eradication in screening programs to reduce the total number of GC cases without delaying the diagnosis.